# Visualizing the Knowledge Domain in Urban Soundscape: A Scientometric Analysis Based on CiteSpace

**DOI:** 10.3390/ijerph192113912

**Published:** 2022-10-26

**Authors:** Jiaxi Yang, Hong Lu

**Affiliations:** 1Digital City Research Center, Wuhan University, Wuhan 430072, China; 2School of Urban Design, Wuhan University, Wuhan 430072, China

**Keywords:** soundscape, CiteSpace, research hotspot, scientometric

## Abstract

The purpose of this study was to identify the main research themes and knowledge structures in the field of urban soundscape. With the continuous expansion of research work in the field of urban soundscape, it has become necessary to carry out a systematic analysis. CiteSpace was used to conduct an information visualization analysis of high-quality literature related to urban soundscape research in the WoS database from 1976 to 2021. The results revealed the following: (1) In terms of research content, research hotspots center on noise, perception, and quality, while focusing on theory and methodology. (2) In terms of research methods, the Perceptual Restorative Soundscape Scale has gradually become the main method of soundscape research. With the development of sound acquisition technology and sound simulation technology, the soundscape perception model will undergo an iterative process of updating. (3) In terms of research objects, most of the research focuses on the soundscape of outdoor environments (such as urban parks, tourist attractions, and historical blocks) together with the influences and preferences for different types of soundscapes. The research results can provide reference for research and planning as well as the design practice of urban soundscape.

## 1. Introduction

The International Organization for Standardization (ISO) defines a soundscape as “an acoustic environment that is perceived or experienced and/or understood in context by one or more persons” [1], ‘soundscape’ is different from ‘acoustic environment.’ The former refers to a perceptual construct, the latter to a physical phenomenon. Soundscapes exist through human perception of the acoustic environment [2]. Thompson (2002) described the relationship between the soundscape resource and the human listener as a cultural construction [3]. Soundscape research represents a paradigm shift from noise control policy to a new multidisciplinary approach [4], as it involves not only physical measurements, but also the collaboration of the humanities and social sciences, taking into account the diversity of soundscapes in different countries and cultures, and paying more attention to how people practically experience the acoustic environment.

The importance of soundscape research has been recognized by European government organizations and national funding agencies, and many national research projects related to this field have been carried out in Europe; examples include the Noisefutures network and related Positive Soundscape project (Engineering and Physical Sciences Research Council) funded by the UK EPSRC, the Soundscape Health project supported by the Swedish Foundation for Strategic Environmental Research, and a series of soundscape projects funded by the PREDIT programme (National Research Initiative for Transport Innovations). Researchers from various countries (such as the United States, Canada, Japan, China, etc.) have become very involved in soundscape research. More than 2400 soundscape-related papers have been published in peer-reviewed international journals over the last 20 years [5].

In the past, some scholars summarized soundscape research from different perspectives. In 2016, Jian Kang and Francesco Aletta et al. systematically expounded the basic definition of soundscape and how to collect and apply sound data [6]. In 2018, VOS Viewer software was used to analyze the association between soundscape and wellbeing [7]. Since the concept of soundscape has been put forward, audio-visual interaction has been incorporated in soundscape research, particularly in urban planning and noise control. Based on the Preferred Reporting Items for Systematic Reviews and Meta-Analyses (PRISMA) guidelines, some scholars have evaluated the existing research methods and effects of audio–visual interaction on soundscape evaluation, as well as soundscape design strategies, from both the global and indoor environmental aspects [8,9]. In a narrative literature review, Eleanor Ratcliffe explores how humans perceive and experience natural environments [10]. In 2022, Like Jiang and Abigail Bristow et al. further discuss valuation of soundscape and research methods based on the latest progress in environmental assessment and soundscape research [11]. Alvaro Balderrama extracted 40 articles based on PRISMA to explore the impact of façades on the urban acoustic environment and soundscape [12]. Nearly two years into a global pandemic of coronavirus disease (COVID-19), acoustic has been one of the environmental factors significantly affected by the COVID-19 pandemic. Hasegawa used PRISMA as a basis to filter out 119 articles exploring how COVID-19 is affecting the soundscape and acoustic environment [13]. So far, the review articles in the field of soundscape basically use the meta-analysis method based on PRISMA, and very few articles use visual analysis technology. Previous studies have not reviewed the dynamic evolution of the soundscape field from a more comprehensive perspective.

This article considers soundscape-related articles in the Web of Science database from 1976 to 2021 as the research object. We used the information visualization software CiteSpace to analyze the collected literature quantitatively. The main purpose of this study is to explore the evolution of the frontiers and research hotspots of soundscape research. The specific issues addressed in this paper are as follows: (1) Understand the changes in the number of total publications and citations for soundscape research, as well as its distribution across countries, institutions and journals; (2) Explore author-institution collaboration networks based on soundscape-related literature; (3) Use keyword co-occurrence analysis to study the evolution of research topics in the field of soundscapes; (4) Use cluster analysis to study research hotspots in this field. This is the first time CiteSpace has been used for visual analysis in the field of soundscape. This research can significantly help relevant industry practitioners understand the field of soundscapes from a macro perspective and identify new research fronts to focus on important aspects of future research.

## 2. Methodology

### 2.1. Data Sources

The Web of Science (WoS) is considered the most comprehensive and accurate scientific and technical knowledge mapping citation database and is classified as the most authentic bibliographic indexing tool for global bibliometric survey data collection [14]. WoS contains sufficient data, such as title, author, institution, country, abstract, keywords, references, citations, impact factor, etc. [15,16]. Considering the high coverage of WoS and compatibility with CiteSpace software, Web of Science was selected as the primary data source.

We retrieved a total of 3282 articles using the following search terms with associated meanings: topic = soundscape or urban soundscape; language = English; year = all; document type = articles and reviews. The earliest collection of literature retrieved from WoS core database dates back to 1976. Thus, we selected literature published between 1976 and the present time for analysis.

### 2.2. Research Method

Scientometrics is a method of evaluating research developments and hotspots among researchers, countries, institutions, journals, and documents in a particular field by generating diverse networks. The approach helps researchers uncover domain-relevant findings by providing a network of connections that might otherwise have been overlooked in manual literature reviews.

CiteSpace is a Java-based free software tool, originally developed by Dr. Chaomei Chen from Drexel University, USA. It enables analysis of diverse co-occurrence networks including articles, authors, keywords, and even turning points, research front and focus [17,18], while it is now widely used for visualization and mapping of scientific literature. This research involves scientometric techniques such as co-authorship analysis, co-word analysis, and co-citation analysis, and conducts an in-depth analysis of the current research status of urban soundscapes.

## 3. Results

### 3.1. The Temporal Distribution of Publications and Journals of Soundscape Research

Figure 1 depicts 3282 publications from 1976 to 2021 with an overall rapid upward trend in the number of studies. The growth trend can be divided into four stages, the first stage is from 1976 to 1997, where no more than ten articles were published every year, which is the embryonic stage. The second stage was from 1998 to 2012, and the number of papers rose steadily. The third stage is from 2013 to 2016, the number of articles was 2–3 times the number of articles in the past, and there was an explosive growth. The last stage is 2017–2021, with a steady increase in publication volume.

### 3.2. Research Cooperation Network Analysis

#### 3.2.1. Co-Institution Analysis

To analyze the cooperation of different academic institutions in related research on urban soundscape, the visualized network diagram obtained using CiteSpace is shown in Figure 2. The node size represents the number of articles published by an institution, and the links between nodes represent the cooperation between institutions. From the point of view of nodes and routes, a total of 585 institutions carried out urban soundscape research (N = 585) and cooperated with each other 630 times (E = 630). Most of the institutions participating in the research are universities and colleges. There is a certain cooperative relationship between the University of Sheffield, University College London, Ghent University, and the Chinese Academy of Sciences while Purdue University, Cornell University, Queensland University of Technology, Auckland University form a research group.

Table 1 portrays the 10 institutions ranked by number of publications. “Centrality” can reflect the impact of research objects in the entire field. The greater the centrality, the higher the representativeness of the corresponding research content in the subject area within a certain period [19]. The University of Sheffield ranked first with 76 research papers, followed by UCL and Ghent University with 49 and 41 papers respectively.

#### 3.2.2. Co-Country Analysis

The national or regional cooperative analysis visualization network for urban soundscape research is shown in Figure 3. So far, 154 countries have made academic contributions in this field (N = 154) and have collaborated with each other 692 times (E = 692). Larger nodes indicate more research work being done in the country. In terms of publication volume, the top five countries are the United States (780), the United Kingdom (461), China (281), Italy (208), and Australia (203). They are not only leading countries in this field, but also important bridges for international scientific research cooperation. The research on urban soundscape in the United States started earlier and occupies a core position in this field.

### 3.3. Co-Citation Analysis

Co-citation analysis is an effective method to determine the level of inter-relationships between authors, literature, and journals by building mapping networks and examining areas of scientific research. Co-citation analysis can better quantitatively reflect knowledge bases, research hotspots, and trends.

#### 3.3.1. Journal Co-Citation Analysis

The journal co-citation network provides insight into the impact of individual journals on this area of knowledge. Analyzing the impact of a journal can help readers and researchers quickly access the relevant information. A visual map of the co-citation analysis of journals with 1320 nodes (N = 1320) and 7844 links (E = 7844) was built using CiteSpace, as shown in Figure 4. Each node represents a journal, and the node size represents the co-citation frequency of each journal.

Table 2 portrays the top 10 most-cited journals in the field of urban soundscape research. “The Acoustical Society of America “(1290 times) topped the list, followed by “Applied Acoustics” (864 times), “Landscape and Urban Planning” (673 times), etc. Core journals are cited most frequently, indicating that they have published more in-depth and comprehensive research articles and are often cited by relevant academic circles.

#### 3.3.2. Literature Co-Citation Analysis

Research articles or publications are the basic elements that make up a scientific literature database. Literature co-citation analysis is an effective and convenient method to investigate the knowledge background and evolution of a specific research field. Analysis of co-cited literature helps to explore important academic achievements in the field. Using CiteSpace for scientometric analysis, a co-citation analysis visualization network containing 1391 nodes (N = 1391) and 5465 links (E = 5465) was obtained, as shown in Figure 5. Only articles co-cited more than 47 times along with their lead authors and publication years are highlighted in the web.

Table 3 portrays the 10 most influential articles in the field of urban soundscape, sorted by the number of citations, and the table includes the first author’s name, citation frequency, publication year, and DOI. The literature by Francesco Aletta et al. (2016) was cited the most (119 times), followed by Kang Jian (105 times) and Bryan (76 times).

Francesco Aletta (2016) proposed a conceptual framework for predictive model development in soundscape research, detailing how to create a model of the relationship between the perceptual and physical properties of the acoustic environment. These models will provide guidance on creating soundscapes in urban planning and design, creating urban environments of high acoustic quality [20]. Jian Kang (2016) summarized ten major issues about the soundscape of the built environment, involving various aspects such as the definition of soundscape, soundscape framework, and application practice [6]. These two articles have certain guiding significance in urban soundscape research.

Bryan C. Pijanowski (2011) proposed a new research field—soundscape ecology—emphasizing the ecological characteristics of sound and its spatiotemporal patterns when it emerges from the landscape. In the future, more attention should be paid to the ecological significance of sound in the landscape [21]. Jerome Sueur (2015) elaborated on the theories and methods used in eco-acoustics research, which can provide a new perspective for the spatial ecology of acoustically sensitive materials in the future [22]. Nathan D. Merchant (2015) described the use of passive acoustic monitoring (PAM) technology [23]. Susan Fuller (2015) used the Ecological Conditions Framework to assess landscape characteristics to explore the intrinsic relationship between landscape characteristics and ecological conditions in forest sites in southeastern Australia [24]. These four papers highlight a topic that many scholars have focused on in recent years—soundscape ecology—indicating that future research will pay more attention to the impact of acoustics on ecosystems.

William J. Davies (2013) gave a detailed introduction to the project “the Positive Soundscape”, discussing the field of soundscape perception from different methods and results [25]. The acoustic or sound environment of an urban space becomes a major factor affecting the perception of an individual’s soundscape [26]. There are many indoor environmental factors that affect people’s living space experience, including thermal comfort, visual comfort, air quality, and acoustic comfort, which have a great impact on residents’ physical and mental health [27]. Especially for soundscape studies, the acoustic factor is one of the most important environmental comfort parameters to consider when designing a space while assessing the quality of its environment [28]. There are two main elements in assessing soundscape perception in any space; the environment containing the collective acoustics and the person perceiving the acoustical environment in a given context [29]. In the future, cultural and social factors should be considered as part of soundscape evaluation research.

Sydney A. Harris (2016) explored the relationship between traditional and eco-acoustic indices in temperate coral reef ecosystems to assess the utility of eco-acoustic indices as a proxy for species assemblage diversity [30]. Jerome Sueur (2014) used the acoustic index of soundscapes to assess and monitor biodiversity [31]. Graeme Shannon (2016) provided guidance to natural resource managers when assessing anthropogenic impacts or formulating conservation policies by reviewing the extensive literature showing that anthropogenic noise is detrimental to wildlife and natural ecosystems [32]. These three papers focus on exploring the relationship between acoustics and biodiversity. Therefore, it is necessary to conduct in-depth research in this field in the future to strengthen the protection of biodiversity in soundscapes.

#### 3.3.3. Author Co-Citation Analysis

Author co-citation analysis is an efficient way to identify the most influential and active authors in a particular scientific research field and explore the distribution of the most cited prominent authors in that field. The author co-citation analysis was carried out on urban soundscape-related research, and the resulting network is shown in Figure 6. Here, each node represents an author, the node size represents the author’s co-citation count, and the dense lines between nodes reveal the relatively close relationship between the top co-cited authors. In addition, Table 4 portrays the top 10 most cited authors, among which R. Murray Schafe has the highest co-citation frequency with 590 times, followed by Bryan C. Pijanowski and Jian Kang with 385 and 363 times, respectively.

### 3.4. Keyword Co-Occurrence Analysis

Keywords summarize some very valuable information about the research topic and the core content of the research article. In addition, different keywords are popular in different periods, and the research trend can be revealed through the evolution analysis of keywords [33]. By analyzing the temporal changes of the centrality and frequency of co-occurring keywords to explore the most important research areas in urban soundscape related literature, a network graph with 503 nodes and 722 links was generated, as shown in Figure 7. 

The network map revealed that the most frequently used keyword is “noise” (313 times). On the one hand, although soundscape started to become a research field in the late 1960s, it only received great attention from researchers, mainly in the field of community noise and environmental acoustics, around 2016 [6]. On the other hand, it shows that the soundscape field has undergone a paradigm shift. How to manage masking unwanted with wanted sounds as well as reducing unwanted sounds is a subject of widespread interest in the soundscape field. In urban environments, natural soundscapes are widely used to improve the quality of the acoustic environment and mask urban noise. Birdsong can mitigate the negative effects of traffic noise [34], Carlos Iglesias Merchan (2014) investigated the impact of national park soundscape characteristics and noise pollution on the perceived impact of park visitors, and provided an economic assessment of noise reduction programs [35]. Natural sounds have restorative and positive effects on emotional, psychological, and physical arousal and cognition. Participants who listened to familiar natural sounds showed better attention and increased sympathetic activity than those who listened to artificial sounds [36].

The other top ten keywords were “soundscape”, “sound”, “environment”, “perception”, “quality”, “exposure”, “landscape”, “model” and “biodiversity”, reflecting urban soundscapes Research in the field focuses on human perception of sound in the environment, how to build more effective models to test how the human body and brain respond to sound, and the use of acoustics to monitor biodiversity.

Figure 8 reflects the top 10 keywords with the highest burst intensity. The word “annoyance” has the highest burst intensity, noise causes annoyance to people and even affects health. Evaluation of noise and noise reduction is a major focus in urban soundscape research. Among the researchers, A.L.Brown (2011) summarized their experience with the assessment of soundscapes through noise disturbance questionnaires [2]. Karina Mary de Paiva Vianna (2015) assessed the impact of noise exposure in six urban soundscapes to elucidate the relationship between urban noise exposure and health [37]. The terms “music”, “aircraft noise”, and “land use” have high bursts, starting and ending in 1996–2015. From 2006 to 2020, scholars tended to conduct research on “soundscape orientation”. Hans Slabbekoorn (2008) introduced the emerging research field of “soundscape orientation”. Soundscape investigation has great research value in future animal orientation research [38].

Keyword time zone maps can be cited in the form of timelines and reveal the historical trajectory, which can be used to analyze research hotspots and emerging research directions in different periods. The connection stands for the correlation between keywords, and from left to right stands for the temporal evolution from 1990 to 2021. According to Figure 9, the keywords over the past five years are “indoor soundscape”, “acoustic ambient noise”, and “soundscape index”, etc. The soundscape perception emotional quality model developed in the outdoor environment is tending to mature, and the focus of future scholars on soundscape research will gradually shift from outdoor to indoor, The model proposed by Simone Torresin (2020) identifies the dimensions of acoustic perception in residential interiors, investigates people’s emotional responses to indoor soundscapes in residential buildings, and provides guidance for indoor soundscape research and practice [39].

### 3.5. Keyword Cluster Analysis

Cluster analysis classifies the entire knowledge domain of urban soundscapes based on co-occurring keywords or co-cited literature. Cluster analysis classifies the collected bibliometric information into distinct clusters by extracting noun terms from the abstract, title, or keywords of a document. A cluster analysis based on co-cited literature was performed on urban soundscape related studies using Citespace, as shown in Figure 10, and a total of 10 clusters was obtained using the LLR algorithm. 

The details of the clusters and their LLR-based title keywords, size, and silhouette index are listed in Table 5. Silhouette is an index to measure the homogeneity of the entire cluster members, the larger the value, the higher the similarity of the cluster members. As can be seen from Table 5, the silhouette index of all clusters exceeds 0.75, which implies that the clustering is homogeneous and highly reliable. The three keywords “exposure”, “annoyance”, and “sound production” constitute the top three largest clusters, indicating that scholars have conducted many studies around these topics, while the smallest cluster is “impact”. 

The five largest clusters are as follows:

#0 Sound is an inevitable part of everyday life, and soundscapes affect human health and quality of life. Since Kaplan and Ulrich proposed the theories of attention recovery and stress recovery, research on how people respond to exposure to soundscapes has never stopped. Stress recovery theory suggests that human interaction with nature can reduce stress. By transforming the acoustic environment of Plaza Bilbao, Karmele changed its sound ambience and quality, the result increased the comfort and pleasure of residents in the square, indicating that a comfortable soundscape environment is conducive to people’s emotional recovery [40]. In the work environment, the more types of noise content there are, the lower the satisfaction and willingness of office workers to work [41]. In certain environments such as hospitals, the troubles caused by sound are closely related to visual fatigue and mental fatigue. Improving the sound environment is conducive to the rest, sleep, and mood of hospitalized patients, and can improve the work efficiency of hospital staff [42]. Exposure to pleasant soundscapes will help you recover from stress more quickly than from unpleasant soundscapes.

#1 We all know that noise can cause people annoyance. How to reduce the problem caused by noise is a topic of constant research. The acoustic environment can have negative or positive effects on human health and mood. Locations where acoustic stressors (i.e., noise) are present tend to elicit negative emotions and provoke avoidance responses (the so-called defensive motivational system), whereas locations without such stressors may elicit positive emotions and elicit approach responses (appetite motivational systems) [43]. A growing body of research shows that unpleasant soundscapes can cause annoyance or sleep disruption, while positively rated soundscapes can be rejuvenating.

#2 The term sound production appears frequently in the field of bioacoustics. There are differences in the mechanisms of sound production in different organisms [44], and in most animals studied for sound production, males have historically been the main sex studied, and they are thought to be the group that primarily uses sound signals [45]. Bussmann’s findings underscore the potential and importance of future research on vocal communication in neglected taxa and in both sexes [46].

#3 Soundscape descriptors are mainly used as a measure of human subjective perception. The perceptual affective quality model proposed by Axelsson, Nilsson [20] is recommended as a soundscape descriptor by previous research and the soundscape protocol ISO 12913-2 [47]. Some researchers studied the relationship between soundscape descriptors and developed a circumplex model in which pleasantness and eventfulness are the two main dimensions, while calmness and vitality are the two alternative dimensions [48,49]. However, some other literature studies found different conclusions about the circumplex model. Aletta and Kang [50] and Hall, Irwin [51] proposed that vibrancy correlates with eventfulness but not pleasantness. There are differences between soundscape descriptors in different domains, and it is necessary to further explore the relationship in the future.

#4 A geographic information system is a tool for modeling the spatial patterns of anthropogenic noise propagation in natural ecosystems. Modeling the spatial patterns and temporal dynamics of noise disturbance can help ecologists better understand and predict how anthropogenic noise affects species and ecosystems at the landscape level [21]. Several studies proposed a geographic information system based on a spatial analysis model approach to identify and map areas susceptible to noise pollution [52]. Kristy E. Primeau developed a soundshed analysis GIS tool to simulate sound in landscapes, further integrating phenomenological methods into landscape studies, allowing researchers to explore how people hear the wider environment [53]. Implementing model computations in GIS can serve as an effective way to analyze soundscapes in different regions, helping to integrate acoustic data with other spatiotemporal environmental information in environmental research and decision-making [54].

#5 Anthropogenic noise has become an important environmental concern due to its wide-ranging effects on animal physiology, behavior, and ecology [32]. Noise from growing transportation networks and human activities related to economic development has become commonplace. The acoustic properties of anthropogenic noise are different from environmental noise (e.g., sounds produced by wind, rain, homogenous, xenobiotic). The ways in which animal behavior is affected can be extensive and complex [55]. Noise from anthropogenic sources can mask important acoustic signals and behaviors, alter hearing thresholds, reduce hearing ability, and have effects on behavior, hormones, and blood pressure in some species [56]. Several studies have highlighted the importance of taking action to reduce the amount and intensity of anthropogenic noise to improve the quality of life or maintain healthy animal populations [57,58].

#6 Soundscapes are regarded as an ecological resource and are used by various wildlife types. These natural sounds in the landscape essentially represent the functional components of the ecosystem and altering the species composition and/or soundscape by changing the acoustic properties may affect the functioning of the ecosystem [59]. Pijanowski et al. (2011) further elaborated on the connection between sounds and landscapes by defining soundscape ecology as all “biological, geophysical, and anthropogenic sounds that emanate from a landscape and which vary over space and time reflecting important ecosystem processes and human activities ” [60]. Some researchers use acoustic signals from biological sounds as a proxy for biodiversity estimation [61]. Assessing acoustic diversity both spatially and temporally aids landscape and biodiversity conservation efforts.

#7 Space can be identified as a collection of shapes, forms, colors, and appearances, mainly appreciated by sight. Space is one of the four components of soundscape, and the influencing factors of soundscape perception are also closely related to spatial experience. Aburawis, AAM proposed a POE tool to construct the relationship between soundscape perception and spatial experience, which is beneficial for indoor soundscape researchers to study and plan future projects [62].

#8 Soundscape perception is an area of great interest. The way humans connect with nature through soundscapes has evolved over time, and human perception of soundscapes is susceptible to the diversity and complexity of the physical environment, which is extremely subjective in nature. The audience’s physical state (age, gender, degree of visual and auditory impairment, behavior) and psychological level (memory, thoughts, attitudes, values, preferences) affect the perception of soundscapes. Women are more likely to or more actively perceive soundscapes than men, and people with natural occupations and older people are more likely to be close to natural soundscapes [63]. In 2013, Payne developed an evaluation system for restorative soundscapes [64], which combined psychological and contextual factors based on the Perceived Environmental Restorativeness Scale (PRS), the Attention Restoration Theory (ART), and the Perceived Restorativeness Soundscape Scale (PRSS). It has a total of 19 items, including six parts, namely attractiveness, proximity, distance, compatibility, coherence, range. The Perceived Restorativeness Soundscape Scale has been widely used in soundscape research. At present, artificial intelligence is developing explosively, and emerging technologies will bring cutting-edge technical methods and means for soundscape perception.

#9 How different soundscapes affect society, people, animals, etc., is an eternal theme. As one of the important environmental factors, soundscape plays an important role in environmental restoration. When evaluating soundscapes, natural soundscapes are often considered pleasant and relaxing. Voices with typical regional cultural values give cultural phenomena and cultural landscapes an emotional dimension, and the emotional characteristics of voices can shape regional cultural characteristics and build the identity of local residents. Sound can stimulate people’s individual perception, form the social and cultural connection of soundscape, and thus construct their spatial perception [65].

## 4. Discussion

### 4.1. The Knowledge Structure of Urban Soundscapes

On the basis of co-citation analysis and cluster analysis, the relationship between keywords, clusters and research topics was sorted out. The knowledge structure of urban soundscape research is shown in Figure 11. First, 30 high-frequency co-cited keywords were extracted as the knowledge base, then the 10 largest clusters obtained by co-citation analysis were used as the main knowledge areas, and finally eight main research topics were extracted from the clusters, including noise Exposure, perception, assessment, modelling, ecology, quality, positioning, and management. It is evident from the knowledge structure that noise exposure is the main research topic in urban soundscape research. People’s exposure to noise can damage their health and affect their emotions, so many scholars study the impact of different noises on the human body. Job et al. (2001) confirmed the human impact of soundscapes (traffic noise, etc.) and environmental landscapes (community parks, playgrounds, etc.) in a socio-acoustics survey near Sydney airport [66]. As a special group, children are more susceptible to the influence of the acoustic environment when their bodies are not yet mature. Some studies have found that children can recover from cognitive fatigue more quickly after being exposed to the sound of fountains and water by simulating different classroom soundscapes. Serious environmental noise is not conducive to children’s continuous concentration [67].

Soundscape perception is another topic of concern, an important variable of which is how people perceive different sounds, and a range of approaches has been used to establish classifications of sounds and soundscapes. In 1996, Hartige developed the Perceived Environmental Restoration Scale (PRS), which described recovery using four characteristics (attractiveness, distance, compatibility, and degree). In 2010, researchers at the Swedish University of Agricultural Sciences developed eight perceptual sensory dimensions [68], namely tranquility, nature, species diversity, space, outlook, shelter, social and cultural. Neil Bruce et al. used a combination of sound walks, semi-structured interviews, and soundscape simulations to study the impact of expectations on soundscape perception and found that people have strong high expectations and inclusiveness [69]. Positive associations arise when people think they can control the source or element of the sound, and negative feelings arise when they can not, while disturbing sounds are more likely to be noticed.

Soundscape management is a part that cannot be ignored. Future urbanization is expected to continue until 2050, with the urban population increasing to 6.5 billion. Based on this trend, urban environmental planning is a dynamic process necessary to ensure the sustainable use of land and other resources to meet the needs of present and future generations. The acoustic environment is an important topic in urban planning, and a good soundscape helps to improve the perceived quality of the acoustic environment and plays an important role in improving and promoting health. On the one hand, urban planners and policy makers should maximize the restoration potential of urban green space, rationally use sound principles and technical means, and use sound masking, shielding, and absorption methods to reduce noise pollution and promote the diversity of birds in parks. On the other hand, artificial means are used to reduce or remove unnecessary, uncoordinated, and annoying sounds in the environment, and planned construction and the use of new materials are used to reduce and remove noise to create a comfortable soundscape.

The currently established knowledge structure can be used as a reference to predict the future development of urban soundscape related research.

### 4.2. Future Research Directions

(1) Optimize, improve assessment, and apply soundscape methods. The ISO 12913 series on soundscapes was first published in 2014, and since then it has been developed to unify methods for defining, standardizing data collection, and reporting requirements, as well as data analysis and its interpretation. Sound signs are signs that constitute the acoustic characteristics of the environment. In the future, the soundscape quality should be evaluated by considering the sound signs and their perceptual effects in the soundscape, and the acoustic comfort of the area should be restored.

(2) Explore people’s preferences for soundscapes in the post-pandemic era. In the epidemic and post-epidemic era, changes in soundscape preferences in general landscape spaces have become an important factor to be considered when designing public spaces. Certain sounds such as throat clearing and coughing may negatively affect people’s psychological perception.

(3) Focus on the construction of soundscapes for disadvantaged groups. Teenagers and the elderly like to participate in outdoor activities and become close to nature. Although people with disabilities do not tend to go out much, sound is good for soothing emotions. How to build a soundscape environment with both safety and aesthetics to bring better soundscape experience to disadvantaged groups can become a future research direction.

(4) In-depth study of the internal relationship between soundscape and ecology and the factors that affect the relationship. Soundscape research plays an important role in biodiversity conservation, and the complexity of soundscapes is linked to ecological integrity. In the future, acoustic complexity monitoring may become an effective tool for landscape management, conservation, and planning, helping to identify potential hotspots of biodiversity and spatial changes in their distribution.

## 5. Conclusions

This paper is the first to conduct a scientometric analysis of the urban soundscape research field to identify the main research themes and knowledge structures in this field. Co-collaboration, co-citation, and cluster analysis were performed on 3282 papers published between 1976 and 2021 (end of November) using CiteSpace. 

Through the above analysis, the following conclusions were drawn: (1) Over the past 40 years, soundscape research has received extensive attention, and urban soundscapes have experienced tremendous development with regard to people’s perception and preference for various types of soundscapes, model building, soundscape evaluation, and ecology. (2) The most influential institutions are the University of Sheffield, University College London, and Ghent University. The United States (780) and the United Kingdom (461) published the most papers among developed countries, and China (281) published the most among developing countries. (3) Keyword analysis shows that urban soundscape research pays much attention to noise, environment, perception, soundscape quality, and impact on people. (4) In terms of research hotspots, “annoyance” has the strongest outbreak degree, and “management” has the longest outbreak time. The most prominent knowledge clusters are exposure, annotation, sound production, and descriptor. 

This study clarifies the main contributing countries and institutions involved in urban soundscape research, reveals high-frequency keywords and research themes, and proposes possible future research directions, which will provide reference and guidance for researchers interested in this field. This study is not without limitations. We chose to derive data from the core database of WoS, and we only selected English-language articles. However, we believe that visual analysis grounded in a large amount of literature-based data will contribute to this area and bring more attention to soundscape issues. Finally, it is necessary to provide a more comprehensive and precise understanding of urban soundscape research by combining the results of the quantitative data analysis with the reading of the literature content to help researchers better grasp the topics, developments, and frontiers in the field.

## Figures and Tables

**Figure 1 ijerph-19-13912-f001:**
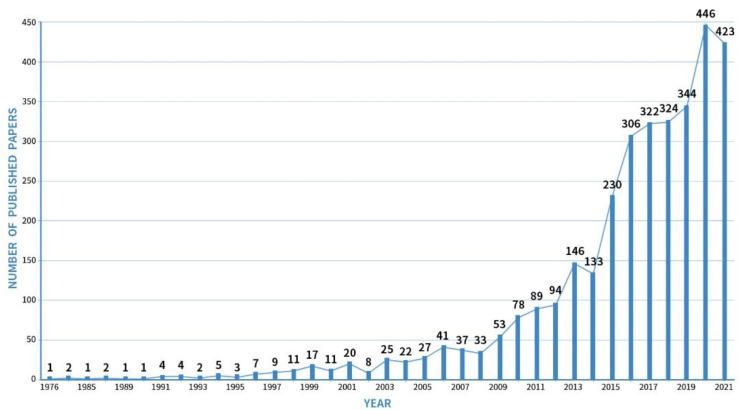
The annual number of soundscape studies published from 1976 to 2021.

**Figure 2 ijerph-19-13912-f002:**
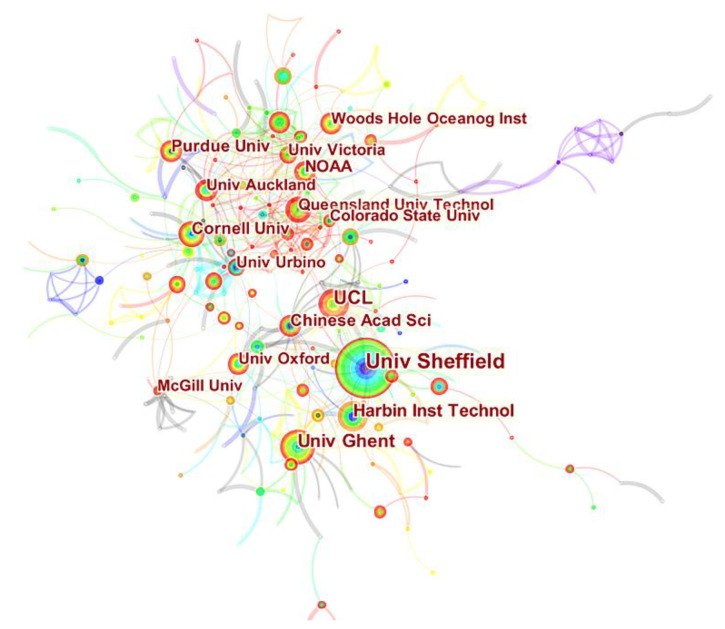
A network map showing institutional collaborations in soundscape research.

**Figure 3 ijerph-19-13912-f003:**
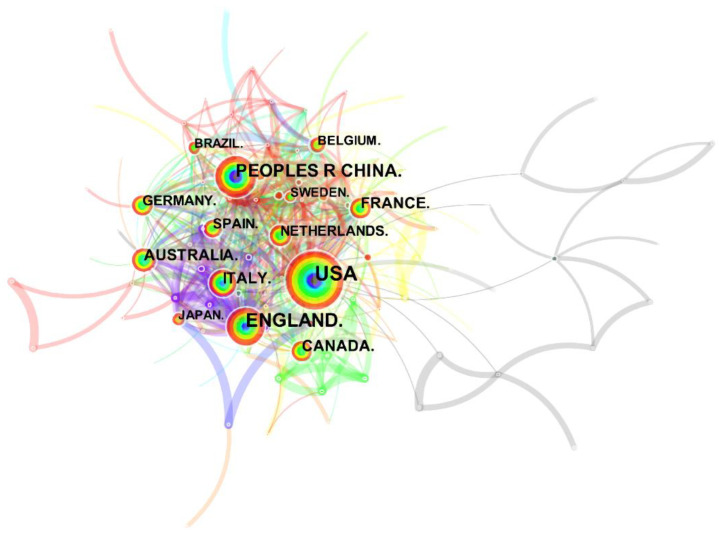
A network map showing national collaborations in soundscape research.

**Figure 4 ijerph-19-13912-f004:**
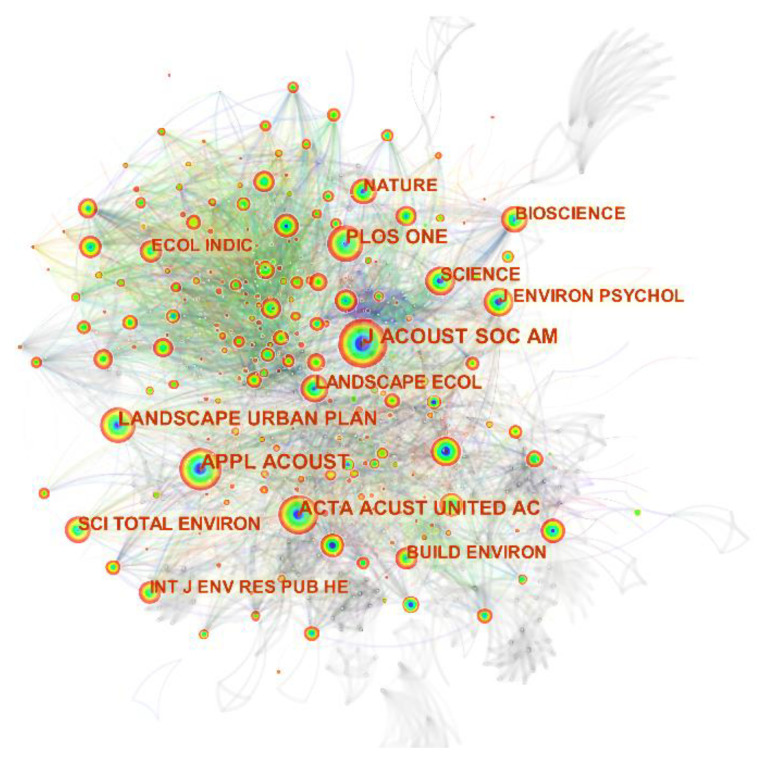
A network map showing journal co-citation in soundscape research.

**Figure 5 ijerph-19-13912-f005:**
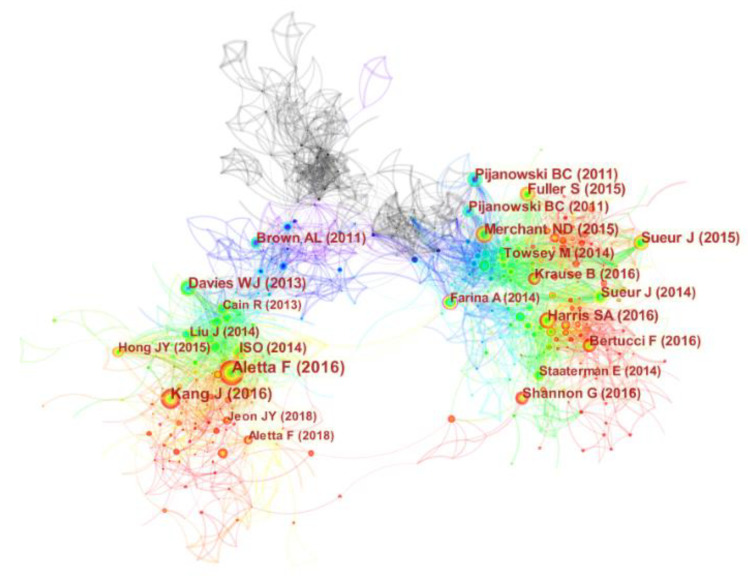
A network map showing literature co-citation in soundscape research [6,20,21,22,23,24,25,26,27,28].

**Figure 6 ijerph-19-13912-f006:**
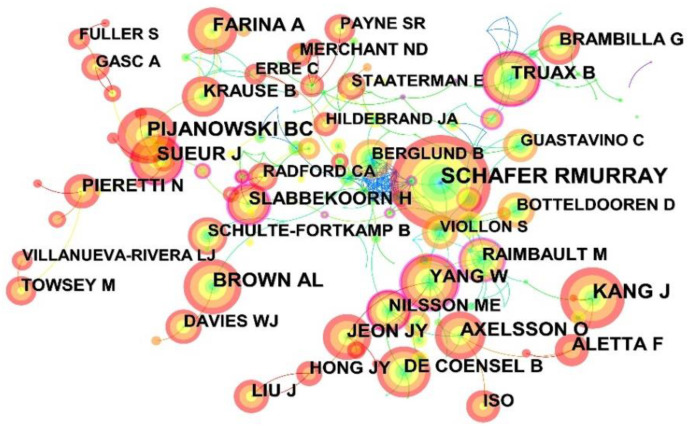
A network map showing authors co-citation in soundscape research.

**Figure 7 ijerph-19-13912-f007:**
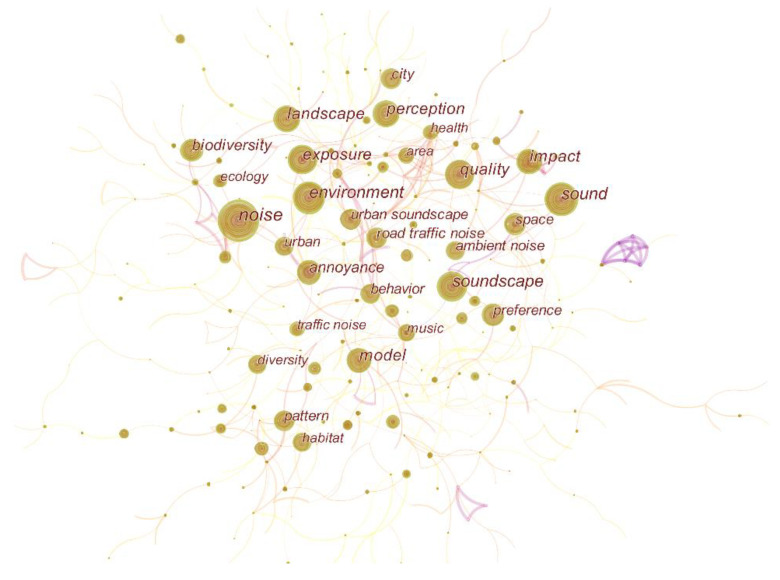
A network map showing keyword co-occurrence in soundscape research.

**Figure 8 ijerph-19-13912-f008:**
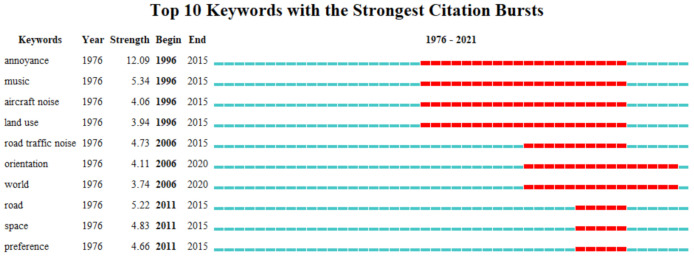
Top 10 keywords with the strongest citation bursts of soundscape research from 1976 to 2021.

**Figure 9 ijerph-19-13912-f009:**
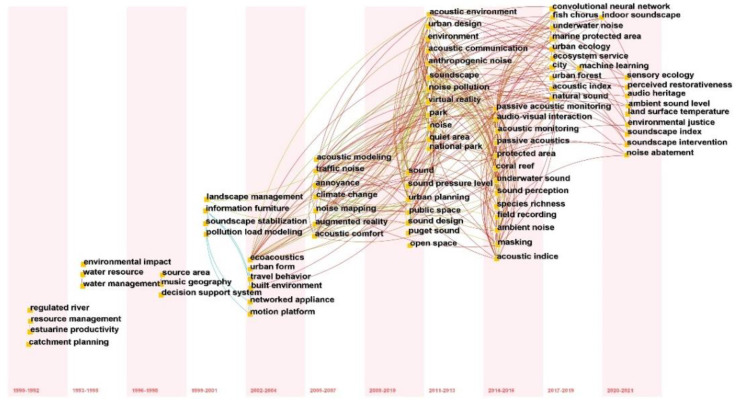
Keywords time zone diagram (timeline view) in soundscape research.

**Figure 10 ijerph-19-13912-f010:**
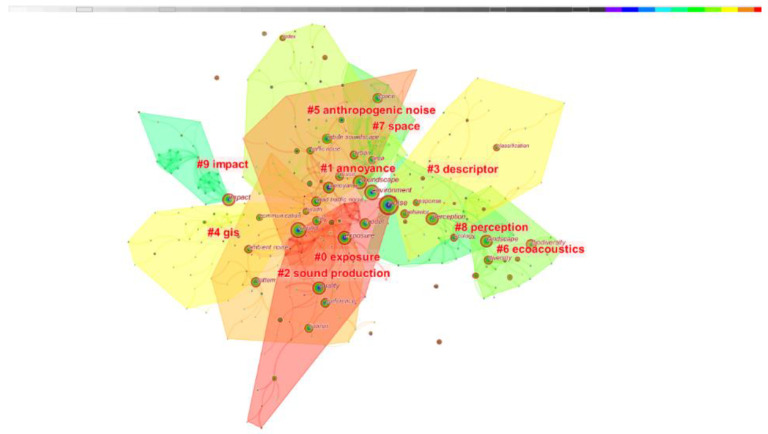
Clustering map of keyword co-occurrence in soundscape research.

**Figure 11 ijerph-19-13912-f011:**
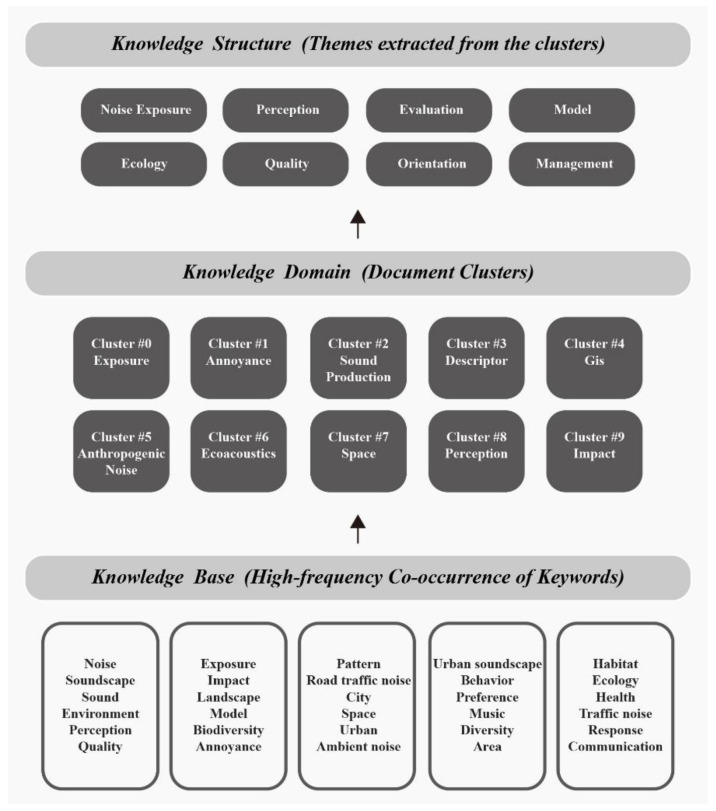
Knowledge structure diagram of urban soundscape research.

**Table 1 ijerph-19-13912-t001:** Top 10 research institutions in the field of soundscape.

No.	Quantity	Institution	Centrality
1	76	The University of Sheffield	0.07
2	49	University College London	0.05
3	41	Ghent University	0.05
4	37	Harbin Institute of Technology	0.01
5	32	Purdue University	0.03
6	32	Cornell University	0.06
7	30	National Oceanic and Atmospheric Administration	0.05
8	29	Chinese Academy of Sciences	0.08
9	28	Queensland University of Technology	0.01
10	26	The University of Auckland	0.04

**Table 2 ijerph-19-13912-t002:** Top 10 most-cited journals in the field of soundscape.

No.	Cited Journals	Citations Count	Year
1	JOURNAL OF THE ACOUSTICAL SOCIETY OF AMERICA	1290	1999
2	APPLIED ACOUSTICS	864	2003
3	LANDSCAPE AND URBAN PLANNING	673	2003
4	PLOS ONE	670	2011
5	ACTA ACUSTICA UNITED WITH ACUSTICA	666	2005
6	SCIENCE	505	1999
7	SCIENCE OF THE TOTAL ENVIRONMENT	443	1997
8	LANDSCAPE ECOLOGY	420	2006
9	THESIS	414	2013
10	NATURE	384	2001

**Table 3 ijerph-19-13912-t003:** Top 10 most-cited literature studies in the field of soundscape.

No.	Cited Literature	Citations Count	Year	DOI
1	Francesco Aletta et al. [20]	119	2016	https://doi.org/10.1016/j.landurbplan.2016.02.001
2	Jian Kang et al. [6]	105	2016	https://doi.org/10.1016/j.buildenv.2016.08.011
3	Bryan C. Pijanowski et al. [21]	76	2011	https://doi.org/10.1525/bio.2011.61.3.6
4	Jerome Sueur et al. [22]	74	2015	https://doi.org/10.1007/s12304-015-9248-x
5	Nathan D.Merchant et al. [23]	71	2015	https://doi.org/10.1111/2041-210X.12330
6	Susan Fuller et al. [24]	70	2015	https://doi.org/10.1016/j.ecolind.2015.05.057
7	William J.Davies et al. [25]	68	2013	https://doi.org/10.1016/j.apacoust.2012.05.010
8	Sydney A. Harris et al. [26]	67	2016	https://doi.org/10.1111/2041-210X.12527
9	Jerome Sueur et al. [27]	60	2014	https://doi.org/10.3813/AAA.918757
10	Graeme Shannon et al. [28]	57	2016	https://doi.org/10.1111/brv.12207

**Table 4 ijerph-19-13912-t004:** Top 10 most-cited authors in the field of soundscape.

No.	Author	Citations Count	Year
1	R. Murray Schafer	590	1996
2	Bryan C. Pijanowski	385	2011
3	Jian Kang	363	2005
4	A.L.Brown	294	2005
5	Jerome Sueur	284	2011
6	Almo Farina	269	2011
7	Osten Axelsson	254	2006
8	Wonseok Yang	243	2005
9	Jin YongJeon	238	2011
10	Francesco Aletta	230	2017

**Table 5 ijerph-19-13912-t005:** The ten largest clusters sorted by size.

Cluster ID	Cluster Label (LLR)	Size	Silhouette	Mean Year
#0	exposure	52	0.876	2012
#1	annoyance	52	0.899	2011
#2	sound production	43	0.906	2015
#3	descriptor	38	0.975	2013
#4	gis	36	0.964	2010
#5	anthropogenic noise	36	0.901	2014
#6	ecoacoustics	28	0.993	2010
#7	space	27	0.963	2005
#8	perception	26	0.869	2009
#9	impact	25	0.961	2003

## Data Availability

Not applicable.

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
