# Peer review of "Visualizing the Knowledge Domain in Urban Soundscape: A Scientometric Analysis Based on CiteSpace"

_ijerph, 2022, doi:10.3390/ijerph192113912_

Round 1

Reviewer 1 Report

Ref. Review: IJERPH-1927360

Paper Title: Visualizing the Knowledge Domain in Urban Soundscape: A Scientometric Analysis Based on CiteSpace

Dear Authors and Editor,

Based on the text exposed in the entitled paper: "Visualizing the Knowledge Domain in Urban Soundscape: A Scientometric Analysis Based on CiteSpace", I recommend major revisions before acceptance and publishing in Environmental Research and Public Health.

The work presented in this manuscript shows a good overview of the field of the soundscape. Still, there are some points to clarify and improve.

Please consider the following recommendations for the improvement of this manuscript:

Abstract:

1)      Please inform the aim of this work.

Introduction:

2)      Please highlight in the introductory text that the authors are analysing a mixture of works that use the term soundscape. In the literature review, there are papers about urban soundscape, bioacoustics, ecological soundscape, and maybe even papers related to environmental noise. That’s why there are expressive numbers of works in some countries, like the United States. There probably, most works are related to ecological soundscape and bioacoustics.

3)      Line 25: could you please substitute the comma for a point after the reference [1]?

4)      There are other soundscape literature reviews made before. Have you checked the following papers? It is important to mention in the introduction the previous works done showing systematic reviews, highlighting the differentiation of this work. By searching on Google Scholar, I found the following papers regarding systematic review on the topic soundscape:

                     • Hasegawa, Y., & Lau, S. K. (2022). A qualitative and quantitative synthesis of the impacts of COVID-19 on soundscapes: A systematic review and meta-analysis. Science of The Total Environment, 157223.

                    • Balderrama, A., Kang, J., Prieto, A., Luna-Navarro, A., Arztmann, D., & Knaack, U. (2022). Effects of Façades on Urban Acoustic Environment and Soundscape: A Systematic Review. Sustainability, 14(15), 9670.

                     • Jiang, L., Bristow, A., Kang, J., Aletta, F., Thomas, R., Notley, H., ... & Nellthorp, J. (2022). Ten questions concerning soundscape valuation. Building and Environment, 109231.

                    • Ratcliffe, E. (2021). Sound and soundscape in restorative natural environments: A narrative literature review. Frontiers in Psychology, 12, 963.

            • Hasegawa, Y., & Lau, S. K. (2021). Audiovisual bimodal and interactive effects for soundscape design of the indoor environments: a systematic review. Sustainability,13(1), 339.

                    • Li, H., & Lau, S. K. (2020). A review of audio-visual interaction on soundscape assessment in urban built environments. Applied acoustics, 166, 107372.

             • Lindseth, A. V., & Lobel, P. S. (2018). Underwater soundscape monitoring and fish bioacoustics: a review. Fishes, 3(3), 36.

                   • Moscoso, P., Peck, M., & Eldridge, A. (2018). Systematic literature review on the association between soundscape and ecological/human wellbeing.

                 • Kang, J., Aletta, F., Gjestland, T. T., Brown, L. A., Botteldooren, D., Schulte-Fortkamp, B., ... & Lavia, L. (2016). Ten questions on the soundscapes of the built environment. Building and environment, 108, 284-294.

Methodology:

5)      Line 63: The citation [6] is too old. Source of 2008, we are in 2022. Could you please update this citation to make this information reliable?

Results:

6)      Figure 1: Could you please improve the quality of this figure?

7)      Figure 2: Interestingly, TU Berlin, RWTH Aachen, University of Salford are not appearing on this network map.

8)      Table 1: The indicated year refers to the year of publication? Or, since this year, have you found publications from the referred university?

9)      Figure 6: Schafer R Murray and Schafer RM are the same person.

10)   Line 227: “The network map revealed that the most frequently used keyword is "noise" (313 227 times),(…).”

Noise is a term used for environmental noise studies, not soundscape. Probably in your literature review, there is a mixture of soundscape and environmental noise papers. Could you please explain why the term noise appeared so frequently?

11)   Lines 245-249: “The other top ten keywords were "soundscape", "sound", "environment", "perception", "quality", "exposure", "landscape", "model" and "biodiversity", reflecting urban soundscapes Research in the field focuses on human perception of sound in the environment, how to build more effective models to test how the human body and brain respond to sound, and the use of acoustics to monitor biodiversity.”

Repetition of text. The previous paragraph has the same content.

12)   Lines 272-273: “(…) outdoor to indoor, The model proposed by Simone Torresin (2020) identifies the dimensions of (..)”

After indoor, substitute the comma for a point, please.

13)   Figure 9: Too small, difficult to read. Could you improve the quality of this figure, please?

14)   Table 5: As far as I know, the Silhouette measure of Cohesion is the value for one clustering model. Here you are showing ten models? Could you please clarify this in the text?

Discussion:

15)   Line 448: Please remove the comma from ISO 12913.

Author Response

Dear Reviewer,

Please see the attachment.Looking forward to your reply.

Reviewer 2 Report

First of all, It has been a discovery in terms of presenting a review of a topic. So I think it is interesting.  

During the reading of the article, I found some little mistakes, or misinterpretations, as you can found in the attached. 

Author Response

Point 1:  redundant.

Response 1: OK, I have edited.

Point 2:  Is it a right word?

Response 2: Sorry, I don't see the label for this word in the attachment.

Point 3:  The.

Response 3: Ok, I have edited.

Point 4:  What does year refer?

Response 4: The year represents the average year of publication, and I now replace the year with the centrality value.

Point 5:   This paragraph is repeated.

Response 5: OK, I've removed duplicates.

Round 2

Reviewer 1 Report

Ref. Review: IJERPH-1927360_rev1

Paper Title: Visualizing the Knowledge Domain in Urban Soundscape: A Scientometric Analysis Based on CiteSpace

Dear Authors and Editor,

Based on the text exposed in the entitled paper: "Visualizing the Knowledge Domain in Urban Soundscape: A Scientometric Analysis Based on CiteSpace", I recommend major revisions before acceptance and publishing in Environmental Research and Public Health.

The work presented in this manuscript shows a good overview of the field of the soundscape. Still, there are some points to clarify and improve.

Please consider the following recommendations for the improvement of this manuscript:

Introduction:

1)      There are other soundscape literature reviews made before. Have you checked the following papers? It is important to mention in the introduction the previous works done showing systematic reviews, highlighting the differentiation of this work. By searching on Google Scholar, I found the following papers regarding systematic review on the topic soundscape:

·        Hasegawa, Y., & Lau, S. K. (2022). A qualitative and quantitative synthesis of the impacts of COVID-19 on soundscapes: A systematic review and meta-analysis. Science of The Total Environment, 157223.

·        Balderrama, A., Kang, J., Prieto, A., Luna-Navarro, A., Arztmann, D., & Knaack, U. (2022). Effects of Façades on Urban Acoustic Environment and Soundscape: A Systematic Review. Sustainability, 14(15), 9670.

·        Jiang, L., Bristow, A., Kang, J., Aletta, F., Thomas, R., Notley, H., ... & Nellthorp, J. (2022). Ten questions concerning soundscape valuation. Building and Environment, 109231.

·        Ratcliffe, E. (2021). Sound and soundscape in restorative natural environments: A narrative literature review. Frontiers in Psychology, 12, 963.

·        Hasegawa, Y., & Lau, S. K. (2021). Audiovisual bimodal and interactive effects for soundscape design of the indoor environments: a systematic review. Sustainability,13(1), 339.

·       Li, H., & Lau, S. K. (2020). A review of audio-visual interaction on soundscape assessment in urban built environments. Applied acoustics, 166, 107372.

·        Lindseth, A. V., & Lobel, P. S. (2018). Underwater soundscape monitoring and fish bioacoustics: a review. Fishes, 3(3), 36.

·        Moscoso, P., Peck, M., & Eldridge, A. (2018). Systematic literature review on the association between soundscape and ecological/human wellbeing.

·        Kang, J., Aletta, F., Gjestland, T. T., Brown, L. A., Botteldooren, D., Schulte-Fortkamp, B., ... & Lavia, L. (2016). Ten questions on the soundscapes of the built environment. Building and environment, 108, 284-294.

Authors‘ answer (Response 4):

“I have read literature reviews on soundscapes, and there are currently no articles that use citespace to explore the interrelationships between articles, authors, and references in the field of soundscapes from a macro perspective”.

Reviewer comment:

Yes, they have not used ‘citescape’, which is one methodology of quantifying the literature review and showing it in clusters. Still, there are other important literature reviews done before the suggested work that should be mentioned. The ‘citescape’ part is the novelty of this work but does not mean that you should ignore years of work and other methodologies made before. I highly recommend that the authors mention the literature reviews realised before, their methodological approach and what they are improving now through ‘citescape’. This will show a sort of evolution in literature review approaches regarding soundscape. In the discussion part, it should be discussed which are the limitations of this approach.

Results:

2)      Table 1: The indicated year refers to the year of publication? Or, since this year, have you found publications from the referred university?

Authors‘ answer (Response 8):

"The year represents the average year of publication, and I now replace the year with the centrality value”.

Reviewer comment:

Please explain to the reader how the centrality measure works. Please add references.

3)      Figure 6: Schafer R Murray and Schafer RM are the same person.

Authors‘ answer (Response 9):

“OK, I have updated this figure”.

Reviewer comment:

Deleting the name ‘Schafer RM’ is not the most proper solution. I still can observe the cluster ‘Schafer RM’ representation in this figure (but without the name identification). Please redo this graph. By adding the quantities of observations of ‘Schafer RM’ into ‘R Murray Schafer’ cluster representation, there will be an evident increase in the cluster ‘R Murray Schafer’ cluster which is not the case in the actual graphical representation. Please consider the reviewers’ comments and avoid this sort of ‘makeup’ that you are showing in this manuscript version.

4      Line 227: “The network map revealed that the most frequently used keyword is "noise" (313 227 times),(…).”

Noise is a term used for environmental noise studies, not soundscape. Probably in your literature review, there is a mixture of soundscape and environmental noise papers. Could you please explain why the term noise appeared so frequently?

Authors‘ answer (Response 10):

“I think that in the field of soundscape, whether it is to discuss people's perception of sound, or the role and impact of sound on people and the environment, noise has always been a part that cannot be ignored. As long as the acoustic environment exists, there will be sounds that affect people or animals”.

Reviewer comment:

Based on ISO 12913-1 the field of soundscape passed through a paradigm shift, where the research should not just focus on noise (unwanted sound). Please observe the concepts stated at ISO 12913-1 and explain in the text why you are also including environmental noise studies in this literature review. The authors can include the investigated papers regarding noise, but they should highlight in the text the paradigm shift, why they included works related to environmental noise and why the term noise appeared so often in this research. Otherwise, this is not a soundscape study.

5      Figure 9: Too small, difficult to read. Could you improve the quality of this figure, please?

Authors’ answer (Response 13):

“This picture mainly shows the ten cluster names calculated by citespace. The text of each cluster cannot be adjusted to be larger, and if it becomes larger, it will block other topics“.

Reviewer comment:

If the figure quality could not be improved, there is no sense in including this figure in the manuscript. Please remove it and find other ways to explain this content.

6      Table 5: As far as I know, the Silhouette measure of Cohesion is the value for one clustering model. Here you are showing ten models? Could you please clarify this in the text?

Authors’ answer (Response 14):

“Yes, figure 10 shows the outline size of ten clusters”.

Reviewer comment:

There is a difference between ten cluster models and ten clusters. One cluster model can show ten clusters. Each model is producing one Silhouette measure of cohesion and separation. So as far as I can observe the authors calculated ten cluster models, but they were not informed how many clusters they have in each model. Please clarify this question properly.

Author Response

Dear Reviewer,

Please refer to the attachment. I really appreciate your suggestions for this article. Now I have revised and improved the article according to your suggestion. Looking forward to your reply.

Sincerely
